# *Halophila stipulacea*: A Comprehensive Review of Its Phytochemical Composition and Pharmacological Activities

**DOI:** 10.3390/biom14080991

**Published:** 2024-08-12

**Authors:** Ziad Chebaro, Joelle Edward Mesmar, Adnan Badran, Ali Al-Sawalmih, Marc Maresca, Elias Baydoun

**Affiliations:** 1Department of Biology, American University of Beirut, Riad El Solh, Beirut 1107 2020, Lebanon; zmc15@mail.aub.edu (Z.C.); jm104@aub.edu.lb (J.E.M.); 2Department of Nutrition, University of Petra, Amman 11196, Jordan; abadran@uop.edu.jo; 3Marine Science Station, University of Jordan, Aqaba 11942, Jordan; a.sawalmih@ju.edu.jo; 4Aix Marseille Univ, CNRS, Centrale Med, ISM2, 13013 Marseille, France

**Keywords:** *Halophila stipulacea*, seagrass, phytochemistry, pharmacology, natural products

## Abstract

*Halophila stipulacea* (Forsskål and Niebuhr) Ascherson is a small marine seagrass that belongs to the Hydrocharitaceae family. It is native to the Red Sea, Persian Gulf, and Indian Ocean and has successfully invaded the Mediterranean and Caribbean Seas. This article summarizes the pharmacological activities and phytochemical content of *H. stipulacea*, along with its botanical and ecological characteristics. Studies have shown that *H. stipulacea* is rich in polyphenols and terpenoids. Additionally, it is rich in proteins, lipids, and carbohydrates, contributing to its nutritional value. Several biological activities are reported by this plant, including antimicrobial, antioxidant, anticancer, anti-inflammatory, anti-metabolic disorders, and anti-osteoclastogenic activities. Further research is needed to validate the efficacy and safety of this plant and to investigate the mechanisms of action underlying the observed effects.

## 1. Introduction

Plants have been widely used as natural remedies in traditional medicine throughout history. In recent years, their use in alternative medicine has been on the rise, mainly due to the perception of plant-based remedies as safer, more affordable, and sustainable options compared to synthetic drugs. Also, scientists have taken an interest in plant-based natural products for drug development due to their natural origin, synergistic effects, and minimal adverse effects. Indeed, most of the current plant-derived drugs are of terrestrial origin. Moreover, marine organisms are considered an untapped source of natural products with more diverse structural and chemical characteristics compared to terrestrial metabolites [1]. Among marine organisms, seagrasses are one of the most overlooked and valuable sources of bioactive compounds with potential therapeutic effects [2].

Seagrasses are marine angiosperms that play a crucial role in supporting the marine ecosystem [3]. They are the only flowering plants that can live underwater, forming extensive meadows that cover up the coastal ocean on every continent except Antarctica [4]. Seagrasses provide essential services to the ecosystem, including serving as habitats, producing oxygen, and sequestering carbon, among others [5,6,7,8]. There are 72 seagrass species that are divided into six major families: Zosteraceae, Hydrocharitaceae, Posidoniaceae, Cymodoceaceae, Ruppiaceae, and Zannichelliaceae [9,10]. Of particular interest to this review is *Halophila* Thouars, which is a diverse seagrass genus that belongs to the Hydrocharitaceae family. Also, with 24 accepted species, it accounts for more than one quarter of all identified seagrass species [11,12]. Species of this genus are found along the tropical and subtropical coastlines in the Indo-West Pacific Ocean [12,13]. The most studied species of this genus are *H. ovalis*, *H. decipien*, and *H. stipulacea,* due to their widespread distribution. This review is particularly focused on *H. stipulacea* (Forsskål and Niebuhr) Ascherson, which is a small marine seagrass native to the Red Sea, the Persian Gulf, and the Indian Ocean [14,15,16]. The seagrass *H. stipulacea* is a Lessepsian migrant that has invaded the Mediterranean Sea as well as the Caribbean Sea, reaching the South American continent [17,18,19].

This review aims to give a comprehensive overview of the phytochemical composition and pharmacological activities of *H. stipulacea*, with the goal of highlighting its therapeutic potential.

## 2. Methods

Articles published between 1970 and 2023 were retrieved through multiple scientific literature databases, including PubMed, Google Scholar, Scopus, ScienceDirect, SciFinder, Dr. Duke’s Phytochemical, and Chemical Abstracts. Google was also used for simple web searches. The literature search was conducted using the following keywords and MeSH terms: “*Halophila stipulacea*”, AND (“phytochemicals”, “bioactive compounds”, “phytochemical content”, “biological properties, or activities, or effects”, “pharmacological properties, or activities, or effects”, “antioxidant”, “anti-inflammatory”, “anticancer”, “antidiabetic”, “lipid-reducing”, “antimicrobial”, “antibacterial”, or “antifungal”).

## 3. Taxonomic Classification of *Halophila stipulacea*

The taxonomic classification of *H. stipulacea* was retrieved from the World’s Register of Marine Species (WoRMS) and is outlined in Table 1.

The seagrass *H. stipulacea* is a small, dioecious, tropical species with pairs of leaves extending from the rhizome via the petiole, which is 3–15 mm long (Figure 1). The leaves are elliptic, oblong, and 3–8 mm wide and 2–6 cm long [15,20]. The base of the leaves is covered by leaf scales that are 2–10 mm wide and 6–18 mm long [21]. Rhizomes are branched, creeping, and have a thick, unbranched root present at each node of the rhizome.

## 4. Ecological Characteristics of *Halophila stipulacea*

The seagrass *H. stipulacea* is native to the Red Sea, Persian Gulf, and Indian Ocean [14,15,16]. However, it has spread outside its native habitats, particularly to the Mediterranean and Caribbean regions, through transportation by commercial and recreational vessels [22,23]. As such, *H. stipulacea* is considered an invasive alien species and is now found along the coastlines of Lebanon, Jordan, Egypt, Saudi Arabia, Turkey, Libya, Tunisia, Cyprus, Greece, Albania, Malta, Italy, Dominica, Venezuela, and India [15,17,18,20,24,25,26,27,28,29,30,31,32,33,34].

The plant *H. stipulacea* has the ability to adapt to a wide range of ecological conditions, including salinity, light intensity, depth, temperature, substrates, and nutrient levels [29,35,36,37,38,39]. Particularly, it is considered a euryhaline species due to its ability to tolerate different salinity levels [39,40]. In fact, *H. stipulacea* can withstand both low and high salinity levels, ranging from 24 to 70 PSU [23,39]. Also, it has the ability to adapt to a wide spectrum of irradiance levels, leading to a high plasticity in its photosynthetic structures, which enables more efficient light absorption [36,37]. Furthermore, *H. stipulacea* typically grows in shallow depths of 1–6 m; however, it can also be found in deeper meadows at 20–70 m and therefore can tolerate a large range of depths [23]. And two unidentified photosynthetic pigments were found in high concentrations in *H. stipulacea*, which may indicate a possible biochemical adaptation to different irradiance levels to optimize growth in deeper areas [41]. It has been reported that *H. stipulacea* also grows across a wide range of temperatures, from 17 to 42 °C [23,38]. Moreover, this seagrass is found in marine sublittoral sediments living on different substrates of sand, silt, mud, and coral rubble [23,42]. Finally, *H. stipulacea* has a limited capacity for the uptake of nitrate, while it has a high capacity and efficiency for the uptake of ammonium. Interestingly, nitrogen fixation by associated diazotrophic epiphytes compensates for the nitrogen limitation of this seagrass under low nitrogen conditions, providing an advantage when competing for resources with other seagrass species [43]. All of these characteristics contribute to the invasive nature of this species, giving it the ability to outcompete other seagrasses.

Furthermore, *H. stipulacea* exhibits notable morphological and biochemical variations in response to different environmental factors, particularly temperature, depth of habitat, and light intensity [29,44,45]. Seasonal variation is observed in the leaf density and characteristics, with a decrease in the number of leaves and an increase in leaf area in winter compared to summer [46]. Also, this seagrass displays maximal productivity during warmer seasons, as evidenced by bigger leaf structures, a higher growth rate, and enhanced storage of energy-rich nutrients like lipids and carbohydrates [43]. Additionally, the length, width, and area of the leaves vary with depth, for example, as depth increases, these leaf descriptors increase, facilitating better light capture [47,48]. Moreover, *H. stipulacea* growing in deeper habitats exhibited higher contents of photosynthetic pigments, including chlorophyll a, chlorophyll b, and carotenoids, thus improving light capture [48,49,50]. The phenol content of *H. stipulacea* leaves was also reported to vary with depth, light, and temperature [14,46,48]. For example, in deeper habitats with little light, the phenol content of *H. stipulacea* leaves was significantly lower [14,48], while a higher phenol content was detected in the winter compared to that in the summer season [46]. The total fatty acid (TFA) content and composition also vary with depth. It was recorded that *H. stipulacea* leaves growing in deeper areas had 25% more TFA than those growing in shallower depths [50]. This increase was mainly associated with a higher content of polyunsaturated fatty acids, which enhances both the fluidity of chloroplast membranes and electron transport in the photosystems, hence optimizing photosynthesis in low-irradiance, deep environments.

## 5. Phytochemical Characteristics of *Halophila stipulacea*

### 5.1. Phytochemical Composition

The invasive seagrass *H. stipulacea* is rich in secondary metabolites, which enable it to withstand various ecological conditions [51,52]. The major groups of phytoconstituents in *H. stipulacea* are polyphenols and terpenoids. Studies on the phytochemical composition of *H. stipulacea* have shown that this seagrass is particularly rich in the flavonoid apigenin [53,54,55,56,57]. Moreover, the glycoterpenoid syphonoside has been identified as one of the main secondary metabolites of *H. stipulacea* [53,56,58,59]. The major phytochemicals found in *H. stipulacea* are summarized in Table 2 and Figure 2.

**Table 2 biomolecules-14-00991-t002:** Phytochemical composition of *Halophila stipulacea* extracts and their major compounds.

Extract Type	Analytical Methods	Main Results	Major Compounds	References
Ethyl acetate fraction of the methanolic crude extract	TLC, UV, HR-MALDI-MS, HPLC	Identification and isolation of 2 phenolic compounds and 2 aromatic organic compounds with potential anticancer effect	- Polyphenols: p-hydroxybenzaldehyde - aromatic organic compounds: bis(2-ethyl hexyl) phthalate, benzoic acid	[55]
Aqueous fraction of the methanolic crude extract	TLC, UV, HR-MALDI-MS, HPLC	Identification and isolation of a nucleotide with potential anticancer effect	- Nucleotide: thymidine	[55]
Diethyl ether fraction of the acetone crude extract	TLC, UV, HR-MALDI-MS, HPLC	Identification and isolation of a flavonoid, a sterol, and 3 fattyacids with potential anticancer effect	- Flavonoid: apigenin - Sterol: stigmasterol - Fatty acids: linoleic methyl ester	[55]
Aqueous crude extract	HPGC	Identification of 7 monosaccharides	- Monosaccharides: mannose, galactose, glucose	[60]
Methanolic crude extract	RP-HPLC	Identification of 11 Phenolic compounds	- Polyphenols: p-hydroxybenzoic acid, caffeic acid, gallic acid, apigenin, apigenin-7-O-glucoside	[60]
Unsaponifiable matter (from the CHCl3 fraction of the MeOH/CHCl3 crude extract)	GC-MS	Identification of 2 non-oxygenated compounds, 6 oxygenated compounds, and 7 sterol compounds	- Non-oxygenated compound: neo-phytadiene - Oxygenated compounds: phytol, β-amyrin (3β-hydroxyl-olean-12-en-3-ol), lupeol (20,29-lupen-3β-ol) - Sterols: β-sitostenone (stigmast-4-en-3-one), β-sitosterol	[60]
Saponifiable matter (from the CHCl3 fraction of the MeOH/CHCl3 crude extract)	GC-MS	Identification of 8 saturated fatty acids and 5 unsaturated fatty acids	- Saturated fatty acids: pentadecanoic acid, palmitic acid - Unsaturated fatty acids: linoleic (9Z,12Z-octadecadienoic) acid, α-linolenic (9Z,12Z,15Z-octadecatrienoic) acid	[60]
Ethanolic extract	GC-MS	Identification of 36 phytochemicals	- Aromatic organic compounds: 2-phenyltridecane, 4-phenyleicosane, 6-phenyltridecane, 2-phenyldodecane	[61]
Ethanolic extract of shoots and roots	RP-HPLC	Identification of 14 phenolic compounds and 10 flavonoids	- Polyphenols: catechein, ferulic, ellagic - Flavonoids: naringin, naringenin, Kaempferol 3-2-p-coumaroylglucose	[57]
Ethanolic extract	GC/MS	Identification of 15 compounds	- butylated hydroxyl toluene, trimethyl-4-hydroxy-2-cyclohex-1-one, hexadecanoic acid (palmetic)	[62]
Ethanolic extract	UPLC-HRMS/MS	Identification of 80 secondary metabolites	- Polyphenols: apigenin, chrysoeriol, cirsimaritin, luteolin, genkwanin	[56]
Hexane extract of leaves	UPLC-HRMS/MS	Identification of 41 phytoconstituents	- Polyphenols: matairesinol	[54]
Ethyl acetate extract of leaves	UPLC-HRMS/MS	Identification of 47 phytoconstituents	- Polyphenols: apigenin	[54]
Methanol extract of leaves	UPLC-HRMS/MS	Identification of 47 phytoconstituents	- Polyphenols: spiraeoside	[54]
Hexane extract of stems	UPLC-HRMS/MS	Identification of 36 phytoconstituents	- Polyphenols: matairesinol	[54]
Ethyl acetate extract of stems	UPLC-HRMS/MS	Identification of 48 phytoconstituents	- Polyphenols: luteolin	[54]
Methanol extract of stems	UPLC-HRMS/MS	Identification of 52 phytoconstituents	- Polyphenols: luteolin, spiraeoside	[54]
Acetone extract	^1^H NMR, ^13^C NMR, and LC-ESIMS	Identification and isolation of a terpenoid	- Terpenoid: syphonoside	[58]
Diethyl ether fraction of the acetoneExtract	^1^H NMR, ^13^C NMR, TLC, and HRESI-MS	Identification and isolation of 2 compounds	- Terpenoid: syphonoside	[59]
Butanol fraction of the acetone extract	TLC, RP-HPLC, and NMR	Identification and isolation of 9 compounds	- Flavonoids: apigenin-7-O-b-(600-O-malonyl-glucopyranoside), apigenin-7-O-b-glucopyranoside - Terpenoid: syphonoside	[53]

**Figure 2 biomolecules-14-00991-f002:**
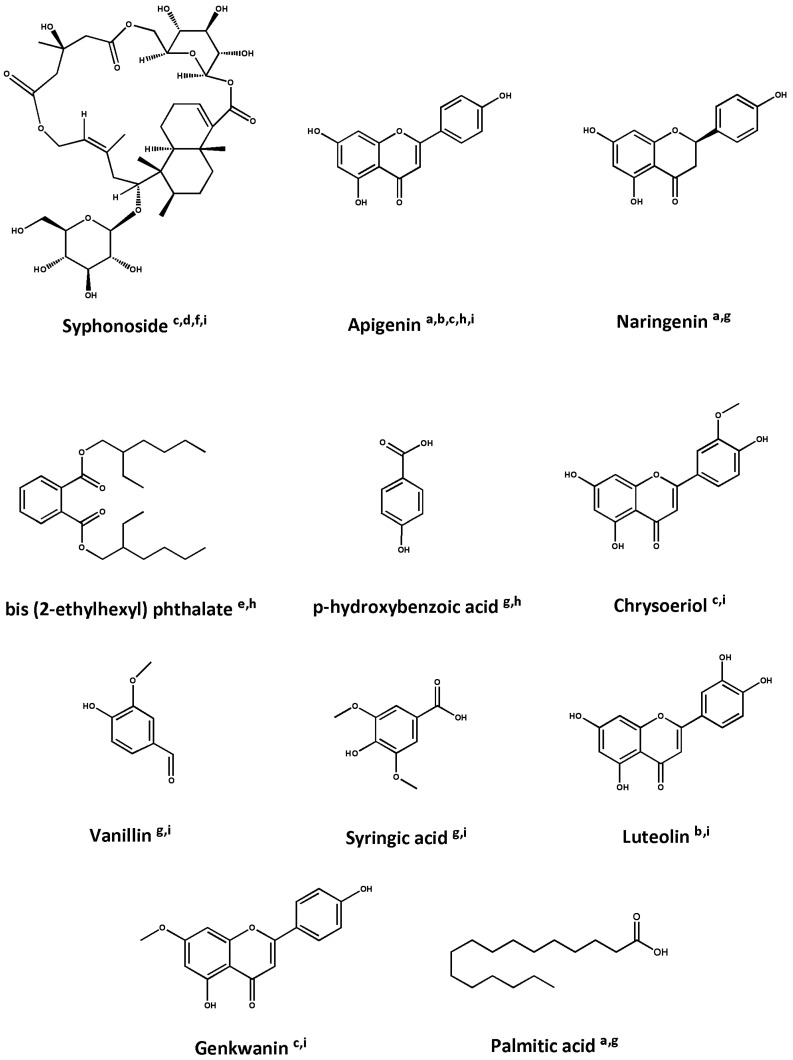
Structures of major metabolites of *Halophila stipulacea.* a: [57], b: [54], c: [53], d: [59], e: [62], f: [58], g: [60], h: [55], and i: [56].

Several extracts of *H. stipulacea* have been shown to be rich in phenolic compounds, flavonoids, and terpenoids. For instance, two phenolic compounds were isolated from the ethyl acetate fraction of the methanolic crude extract, and one flavonoid was identified in the diethyl ether fraction of the acetone crude extract [55]. Similarly, fourteen phenolic compounds and ten flavonoids were identified in an ethanolic extract of shoots and roots [57]. Additionally, eleven phenolic compounds were identified in the methanolic crude extract, including p-hydroxybenzoic acid, caffeic acid, and gallic acid [60]. Moreover, extracts from leaves and stems using different solvents, such as hexane, ethyl acetate, and methanol, were shown to contain various polyphenols such as apigenin, luteolin, and spiraeoside [54]. Also, eighty secondary metabolites, including polyphenols, like apigenin, chrysoeriol, and luteolin, were identified in an ethanolic extract [56]. Furthermore, the glycoterpenoid syphonoside has been identified in the ethanolic extract and acetone extract, as well as the butanol and diethyl ether fractions of the acetone extract [53,56,58,59]. Overall, the variety of extracts from *H. stipulacea* indicate the presence of several bioactive compounds. Phenolic compounds, flavonoids, fatty acids, sterols, and terpenoids are particularly notable. These findings provide evidence of the rich chemical profile of *H. stipulacea* and its potential applications in plant-derived pharmaceuticals.

### 5.2. Nutritional Composition

The nutritional composition of *H. stipulacea* is summarized in Table 3. In a study investigating the biochemical composition of *H. stipulacea*, it was found that the ash, lipid, and protein contents were 14.56 ± 2.08%, 3.16 ± 0.48%, and 8.11 ± 0.07%, respectively [63]. Actually, *H. stipulacea*, with a total protein content of 3.06%, can be considered a good source of essential amino acids, especially valine, threonine, methionine, and leucine [60]. Furthermore, *H. stipulacea* showed considerable amounts of saturated and unsaturated fatty acids, particularly palmitic acid, pentadecanoic acid, linoleic acid, and α-linolenic acid [60,62]. The analysis of the carbohydrates in *H. stipulacea* revealed that galactose, mannose, and glucose were the major monosaccharides [60]. A comparison of the macromolecular composition between young and old leaves of *H. stipulacea* showed that protein, carbohydrate, and lipid concentrations were significantly higher in the young leaves compared to the old leaves [64]. The study showed that the protein, carbohydrate, and lipid concentrations in young leaves were 21.3 ± 0.5, 19.0 ± 0.6, and 5.8 ± 0.1 mg/g dry leaf, respectively, while they were 8.2 ± 0.6, 10.9 ± 1.1, and 1.6 ± 0.3 mg/g dry leaf, respectively, for the old leaves.

In a study examining the metal concentrations in the seagrass *H. stipulacea* across different seasons, it was found that Fe, Zn, and K concentrations were highest in summer and autumn [65]. Also, the leaves of *H. stipulacea* had higher levels of Zn, Na, K, and Mg compared to the stems, roots, and rhizoids [65]. Overall, *H. stipulacea* exhibits significant nutritional value, with young leaves having higher nutritional content compared to old leaves, indicating their potential health benefits.

## 6. Biological Activities of *Halophila stipulacea*

Numerous studies using different solvents for *H. stipulacea* extraction, such as ethanol, methanol, and acetone, have been shown to exhibit various biological activities. These include antimicrobial, antioxidant, anticancer, anti-inflammatory, anti-metabolic disorders, and anti-osteoclastogenic activities, as summarized in Figure 3.

### 6.1. Antimicrobial Activities

The effects of *H. stipulacea* extracts against various microorganisms, including bacteria, fungi, yeast, biofouling bacteria, and biofouling mussel larvae, are summarized in Table 4.

Results showed that ethanolic, chloroform, and ethyl acetate extracts from *H. stipulacea* leaves inhibited all seven investigated bacterial pathogens, especially *Pseudomouos aeruginsa*, with the ethanolic extract showing the highest potency [66]. Additionally, the *H. stipulacea* ethanolic extract of the leaves works synergistically with different antibiotics used for the treatment of *Pseudomouos aeruginsa* [66]. An aqueous extract from *H. stipulacea* leaves showed antibacterial activity only against *Pseudomouos aeruginsa* [66]. Conversely, it was found that an *H. stipulacea* aqueous extract was not active against Gram-negative bacteria, while it showed strong antibacterial activity against the Gram-positive bacteria *Bacillus subtilis* and strong antifungal activity against the filamentous fungi *Aspergillus niger* and the yeast *Candida albicans* [67]. Moreover, the antibacterial activity of *H. stipulacea* leaves was evaluated against seven bacterial strains using three different solvents for extraction. Results showed that the chloroform and hexane extracts were effective against all tested bacteria, with the latter not showing any activity against *Staphylococcus aureus* [68]. An ethanolic extract of *H. stipulacea* showed a strong inhibitory effect on the gram-negative bacterium *Proteus vulgaris*, while it did not show any effect against gram-positive bacteria, other tested gram-negative bacteria, or fungi [61]. Moreover, hexane and ethyl acetate extracts of *H. stipulacea* were tested against different strains of bacteria, yeast, and fungi [69]. The ethyl acetate extract showed an inhibitory effect, particularly against *listeria monocytogenes* and *Salmonella enterica*. However, no significant antifungal effect was shown by all extracts. Interestingly, the antibacterial and antifungal activities of *H. stipulacea* extracts obtained by different solvents were assessed against infectious plant pathogens [70]. Results showed that hexane and chloroform extracts exhibited moderate antibacterial activity against *Pseudomona saeruginosa*, which was higher than other tested seagrasses. Also, both extracts showed strong antifungal effects against *Colletotrichum capsici*.

Furthermore, the antibacterial properties of cotton fabric treated with *H. stipulacea* ethanolic extract were tested against *Staphylococcus aureus* and *Escherichia coli* without washing and after several washing cycles [62]. Results showed that the treated finishing of the fabric inhibited the bacterial growth of both microorganisms by 90% after 10 washing cycles. In addition, *H. stipulacea* leaf and stem extracts obtained with different solvents were evaluated for their anti-biofouling activity against marine bacteria and mussel plantigrade larval settlement [54]. The hexane extract of leaves showed the highest potency against *Halomonas aquamarina* and *Pseudoalteromonas atlantica*, while the ethyl acetate extract of leaves was most active against *Cobetia marina*. Additionally, the hexane extract of stems showed significant inhibition against the settlement of *Mytilus galloprovincialis* plantigrade larvae.

The methanolic extract of *H. stipulacea* showed the most potent antibacterial activity against several bacterial strains, including *Shigella dysentriae*, *Vibrio cholera*, *Staphylococus aureus*, *Micrococcus luteus*, and *Pseudomonas aeruginosa* [68,69,70]. The methanolic extract also exhibited the strongest antifungal activity, particularly against *Macrophomina phaseolina*, *Colletotrichum capsici*, and *Aspergillus flavus* [70]. Moreover, the methanolic extract showed strong anti-biofouling activity by inhibiting the settlement of *Mytilus galloprovincialis* plantigrade larvae, as well as by inhibiting the growth of the marine bacteria *Halomonas aquamarina* and *Cobetia marina* [54]. Overall, methanol can be considered the optimal solvent for extracting bioactive compounds with potent antimicrobial activity from *H. stipulacea*.

### 6.2. Antioxidant Activities

Under normal physiological conditions, there is a balance between the levels of pro-oxidants, such as reactive oxygen species (ROS), and antioxidants [71,72]. Excessive ROS production disrupts this equilibrium, shifting the cell towards the pro-oxidant state, resulting in oxidative stress and cell damage [71,72,73]. Endogenous antioxidants prevent cell damage by managing the level of ROS. Also, antioxidants can be enhanced by exogenous sources to improve cellular defense mechanisms.

The antioxidant activity of different *H. stipulacea* extracts has been investigated using various assays, as shown in Table 5. An ethanolic extract of *H. stipulacea* leaves had the highest level of phenols of all investigated seagrasses, and it showed high total antioxidant activity (75.027 ± 5.199 mg ascorbic acid/g) and good 2,2-diphenyl-1-picrylhydrazyl (DPPH) radical scavenging activity reaching 67.413 ± 0.956%; both being higher than in the six other tested seagrass species [32]. Also, the extract showed the highest reducing power (46.289 ± 1.002 mg gallic acid/g) compared to the other tested seagrasses, according to the ferric ion reducing antioxidant power (FRAP) assay. While a methanolic extract of *H. stipulacea* showed moderate antioxidant activity, reaching 40% DPPH inhibition [67]. Additionally, the ethanolic extract of *H. stipulacea* leaves showed strong DPPH radical scavenging activity (79% inhibition) at low concentrations [66]. Moreover, both an *H. stipulacea* aqueous crude extract and its phenolic fraction showed mild antioxidant activity via the DPPH radical scavenging assay, with a half-maximal inhibitory concentration (IC50) of 13.3 ± 0.25 and 12.6 ± 1.9 µg/mL, respectively [60].

Furthermore, the antioxidant activity of *H. stipulacea* leaves was assessed at different stages of maturity. Specifically, an ethanolic extract of old leaves showed 85% DPPH inhibition at 100 µg/mL, which was greater than the ethanolic extract of young leaves, which showed 45% inhibition at the equivalent concentration [64]. Similar results were obtained using the H_2_O_2_ damage protection assay, where the pre-treatment of WI38 cells with the old leaf extract increased the metabolically active percentage up to 109%, as opposed to 81% when pre-treated with the young leaf extract. Consistent with the previous results, the cell viability recovery assay showed that the metabolically active cell percentage of the injured WI-38 cells increased to 100% when treated with the old leaf extract, while it increased to 83% after the treatment with the young leaf extract. Furthermore, the expression levels of oxidative stress defense genes were analyzed in order to study the effect of the ethanolic extract of *H. stipulacea* mature leaves on the intracellular antioxidant response. Results showed that pre-treatment of WI38 cells with the ethanolic extract of *H. stipulacea* mature leaves before exposure to H_2_O_2_ affected the expression of oxidative stress genes, suggesting that the oxidative stress response was not activated. Moreover, exposing WI38 cells to H_2_O_2_ injury and then treating the cells with the ethanolic extract from mature leaves of *H. stipulacea* significantly upregulated genes associated with the antioxidant response. These findings demonstrate that the extract could have more therapeutic than preventive effects against oxidative damage [64]. It is worth mentioning that the antioxidant potential of *H. stipulacea* has been assessed in vitro. Therefore, further studies are needed regarding its antioxidant capacity and to provide more evidence of its effectiveness in vivo.

### 6.3. Anticancer Activities

Cancer is a malignant disease characterized by uncontrolled cell growth and division. Despite major breakthroughs in cancer treatment, it remains the second-leading cause of death around the world. This indicates the need for an alternative approach, with herbal medicine gaining increasing interest due to its efficacy and few side effects. Figure 4 and Table 6 provide an overview of the anticancer properties of *H. stipulacea* on different cancer cell lines.

The anti-proliferative activity of *H. stipulacea* leaves and stem extracts obtained by different solvents was evaluated against human osteosarcoma MG-63, human neuroblastoma SHSY5Y, and human colorectal carcinoma HCT116 cancer cells, as well as the normal hCMEC cell line [54]. Results showed that both the hexane and the ethyl acetate extracts of leaves and stems showed significant anti-proliferative effects against the tested cancer cell lines. Also, the hexane extract of leaves caused a slight increase in the dead cell count in a 3D cell culture model of human colorectal carcinoma HCT116 cells. However, all extracts decreased the viability of the hCMEC normal cell line and are regarded as generally cytotoxic. In another study, the cytotoxic effect of the aqueous extract of *H. stipulacea*, as well as the chloroform fraction, unsaponifiable matter, and phenolic fraction, was investigated against human ovarian cancer SKOV-3, breast cancer MCF-7, cervical cancer HeLa, prostate cancer DU-145, and pancreatic cancer PANC-1 cell lines. The aqueous crude extract exhibited a strong anticancer effect against prostate DU-145 and pancreatic PANC-1 cancer cell lines [60]. In addition, the chloroform fraction showed strong cytotoxicity against prostate DU-145, cervical HeLa, and pancreatic PANC-1 cancer cell lines. Also, the unsaponifiable matter had a strong anticancer effect, particularly against the human breast cancer MCF-7 cell line. Furthermore, the anticancer properties of different fractions of *H. stipulacea* extracts were tested against several cancer cell lines, including human liver cancer HepG2, colon cancer HCT-116, breast cancer MCF-7, and cervical cancer HeLa cell lines. The diethyl ether fraction showed the strongest anticancer effect against liver cancer HepG2 and breast cancer MCF-7 cell lines, followed by the ethyl acetate fraction, and lastly by the butanol fraction [56]. Conversely, the ethanolic extract of *H. stipulacea* roots and shoots exhibited the lowest anti-proliferative activity compared to the other tested plant extracts against human pancreatic cancer PA1, lung cancer A549, prostate cancer PC3, and colon cancer Caco2 cell lines [58].

The ubiquitin–proteasome system is a complex that is responsible for protein degradation and regulating various cellular processes, including DNA repair and cell proliferation [74,75]. In cancer treatment, targeting the proteasome system is considered a promising therapeutic strategy. In this context, the effect of different fractions of *H. stipulacea* was explored on the ubiquitin–proteasome pathway. It was shown that the proteasome activity was inhibited by the diethyl ether fraction by 97%, followed by the butanol fraction at 86%, the aqueous fraction at 61%, and lastly by the ethyl acetate fraction at 50% [55]. Additionally, the effect of *H. stipulacea* fractions was tested on the interaction of the ubiquitin-conjugating enzyme complex (Uev1A-Ubc13). Results showed that all tested fractions inhibited the Uev1A-Ubc13 interaction, which is associated with the inhibition of apoptosis as well as the promotion of tumorigenesis and metastasis. Also, the effect of *H. stipulacea* fractions on the interaction of P53-Mdm2 was investigated. Mdm2, which is a negative regulator of the p53 tumor suppressor protein, binds to p53 and leads to its degradation. Results indicated that all tested fractions inhibited the P53-Mdm2 interaction, with the ethyl acetate fraction showing the highest inhibition.

In conclusion, these findings provide evidence that *H. stipulacea* exhibits strong anti-proliferative and potential anticancer effects by targeting multiple mechanisms involved in the ubiquitin-proteasome pathway.

### 6.4. Anti-Inflammatory Activity

Inflammation is a natural biological reaction to heal an injury or fight an infection. As inflammation becomes chronic, it could damage healthy cells, leading to several diseases, including heart disease, diabetes, and cancer.

The anti-inflammatory effect of cotton fabrics treated with *H. stipulacea* ethanolic extract was investigated using a carrageenan-induced paw edema model in rats [62]. The treated cotton fabrics showed high anti-inflammatory activity, as observed by the significant decrease in paw edema percentage in the tested rats. Particularly, fabrics treated with *H. stipulacea* ethanolic extract alone showed an anti-inflammatory potency of 54%. Moreover, fabrics treated with microencapsulated *H. stipulacea* ethanolic extract demonstrated higher anti-inflammatory properties. Indeed, *H. stipulacea* ethanolic extract, encapsulated with alginate and mybro, exhibited an anti-inflammatory potency of 69.2 and 71.4%, respectively. Stronger anti-inflammatory effects were observed after post-treating the fabrics with citric acid. Alginate and mybro microencapsulated *H. stipulacea* ethanolic extract post-treated with citric acid showed 88.3 and 83% anti-inflammatory potencies, respectively. These findings suggest that *H. stipulacea* has strong anti-inflammatory effects, warranting further investigation into the mechanisms of action and exploring its full potential.

### 6.5. Anti-Metabolic Disorders Activities

Diabetes mellitus is a chronic metabolic disease that causes the blood sugar level to become high. Over time, this elevated blood sugar damages blood vessels and nerves, which may lead to blindness, kidney failure, and cardiovascular disease. Natural products, such as polyphenols and alkaloids, have been considered a good source of anti-diabetic compounds. These phytochemicals exert their anti-diabetic effect by enhancing the efficacy of pancreatic tissue by inducing insulin secretion and by decreasing glucose levels in the blood [76].

Interestingly, *H. stipulacea* ethanolic extract strongly inhibited the digestive enzymes α-amylase, *β*-glucosidase, and pancreatic lipase, which may be due to the high concentration of phenolic compounds in the extract [56]. Furthermore, the oral administration of *H. stipulacea* ethanolic extract significantly decreased blood sugar levels and increased insulin levels in a dose-dependent manner in a streptozotocin-induced diabetic rat model [56]. Also, the extract caused an increase in the levels of the glucose transporter (GLUT2) as well as the nitric oxide (NO) levels, suggesting that treatment with *H. stipulacea* may increase glucose uptake and enhance endothelial function, further contributing to its antidiabetic effects. In addition, total cholesterol, HDL-cholesterol, and triglyceride levels were significantly decreased. Moreover, the treatment with *H. stipulacea* caused a decrease in the level of malondialdehyde (MDA), which is an indicator of free radical-induced lipid peroxidation; this may be attributed to the antioxidant effect of the extract. However, in another study, extracts prepared from *H. stipulacea* leaves and stems using different solvents did not show any significant effect on the glucose uptake of HepG2 cells [54]. This indicates that the extraction method may influence the antidiabetic activity of this plant, and further studies are needed to determine the mechanisms of action behind its antidiabetic effects.

There is a strong correlation between obesity and diabetes, as excess body weight is a major risk factor for diabetes mellitus [77]. Extracts prepared from *H. stipulacea* leaves and stems using different solvents were tested for their lipid-reducing activity on the zebrafish larvae [54]. Results showed that the ethyl acetate and methanolic leaf extracts caused a significant reduction in Nile red staining after 48 h of exposure, indicating strong lipid-lowering activity. However, these extracts had no anti-steatosis activity, as no significant lipid-reducing effect on the fatty acid-overloaded HepG2 cells was observed [54].

These findings suggest that *H. stipulacea* could be considered as a potential source of anti-diabetic and hypolipidemic compounds for the treatment of diabetes and obesity, requiring further investigation (Table 7) and (Figure 5).

### 6.6. Anti-Osteoclastogenic Activity

Bone remodeling occurs through the removal of mineralized bone by the osteoclasts and the formation of bone matrix by the osteoblasts [78]. Excessive osteoclast activity caused by the activation of osteoclastogenesis leads to several diseases, including osteoporosis, periprosthetic osteolysis, rheumatoid arthritis, Paget’s disease, and osteoclastoma [79,80].

The effect of different fractions of *H. stipulacea* extracts on osteoclastogenesis was investigated using the tartrate-resistant acid phosphatase (TRAP) assay on RAW264 cells (Table 8) [55]. The diethyl ether and the butanol fractions of the acetone crude extract showed the strongest anti-osteoclastogenic effects, reaching an inhibition percentage of 117 and 114%, respectively. The presence of phytosterols, particularly stigmasterol, along with fatty acids in the diethyl ether fraction could possibly be the reason behind the observed strong effect.

### 6.7. Other Activities

Nanotechnology is a fast-growing field with extensive applications across various scientific and technological domains [81,82]. In recent years, there has been an increased demand for the development of a non-toxic, environmentally friendly technology for the production of nanoparticles. Consequently, biological systems have been gaining interest in the green synthesis of nanoparticles, particularly bacteria, fungi, and plant extracts [83,84]. Accordingly, *H. stipulacea* aqueous extract was successfully used in the green synthesis of silver nanoparticles (Ag-NPs) and iron oxide nanoparticles (Fe_3_O_4_-NPs) [85]. The extract was rich in polyphenols and proteins that might be responsible for the reduction of silver and iron ions into nanoparticles. In addition, polysaccharides from the extract resulted in the stabilization of the biosynthesized Ag-NPs and Fe_3_O_4_-NPs. The Fe_3_O_4_-NPs biosynthesized by *H. stipulacea* aqueous extract successfully facilitated the separation of Ag-NPs. Moreover, these nanoparticles were very effective in inhibiting the growth of the cyanobacterium *Oscillatoria simplicissima*, which produces neurotoxins harmful to aquatic organisms, as evidenced by the decreased optical density and total chlorophyll levels. In summary, eco-friendly nanoparticles were successfully synthesized by *H. stipulacea*, showcasing its potential in green nanotechnology.

## 7. Safety of *Halophila stipulacea*

The safety and efficacy of plants with therapeutic effects are mainly based on their long-term traditional use. Nonetheless, plant-derived medicines require assessment of safety, quality, and efficacy before human consumption [86,87].

The toxicity of *H. stipulacea* extracts prepared from leaves and stems using hexane, ethyl acetate, and methanol as solvents was assessed on zebrafish larvae at concentrations of 2 or 6 µg/mL [54]. No mortality was observed among the larvae at these concentrations after 24 and 48 h of treatment. However, additional toxicological screenings and tests are required to verify the safety and efficacy of this plant.

## 8. Conclusions and Future Perspectives

Research on plant-derived natural products has been increasing over the past years, leading to the successful development of drugs used in the treatment of a variety of diseases, showcasing their potential in advancing medical treatments. Studies on the seagrass *Halophila stipulacea* highlight its potential as a rich source of novel bioactive compounds with diverse pharmacological activities. For example, the demonstrated antimicrobial and antioxidant effects of *H. stipulacea* suggest possible applications in the development of natural preservatives. Additionally, the nutritional value of *H. stipulacea*, along with its anti-diabetic and hypolipidemic properties, indicates its potential use in developing natural health-promoting supplements. Moreover, due to its diverse biological effects, including anticancer, anti-inflammatory, and anti-osteoclastogenic effects, *H. stipulacea* is considered a promising candidate for further research. Additional studies on the molecular mechanisms of action and safety profiles of compounds derived from *H. stipulacea* are needed to fully explore the potential of this seagrass.

## Figures and Tables

**Figure 1 biomolecules-14-00991-f001:**
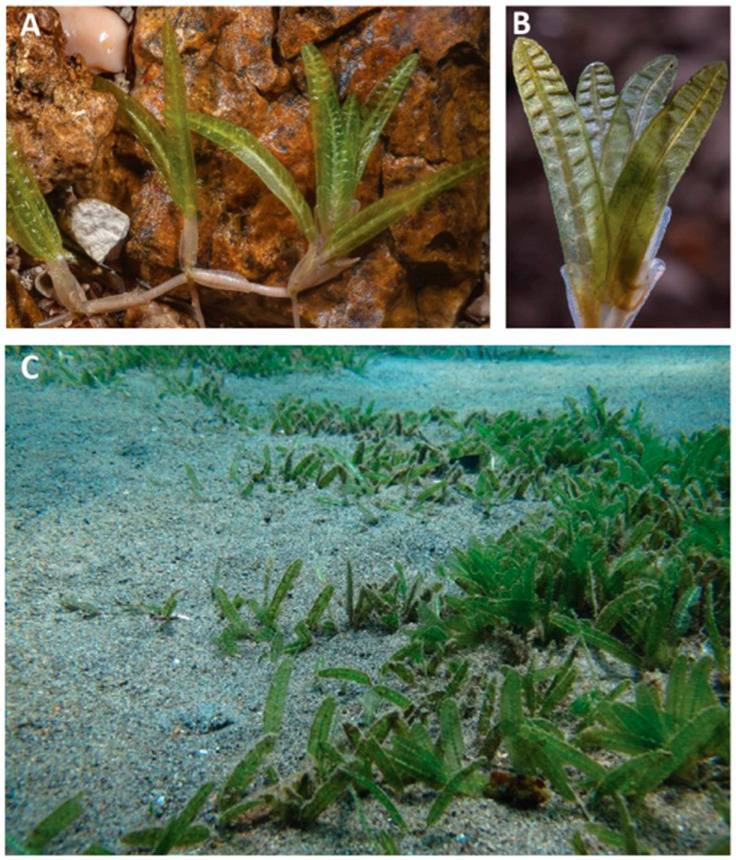
*Halophila stipulacea*. Images were obtained from (**A**,**B**): https://www.floraofqatar.com/halophila_stipulacea.htm (accessed on 1 June 2024), (**C**): https://inpn.mnhn.fr/espece/cd_nom/368620/tab/fiche (accessed on 1 June 2024).

**Figure 3 biomolecules-14-00991-f003:**
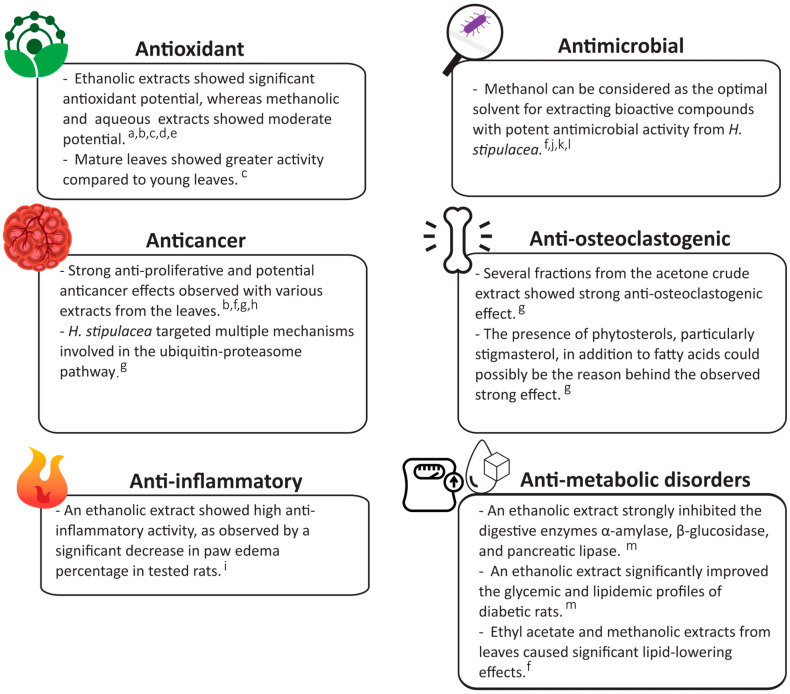
Pharmacological activities of *Halophila stipulacea*. a: [32], b: [60], c: [64], d: [66], e: [67], f: [54], g: [55], h: [57], i: [62], j: [68], k: [69] l; [70], and m: [56].

**Figure 4 biomolecules-14-00991-f004:**
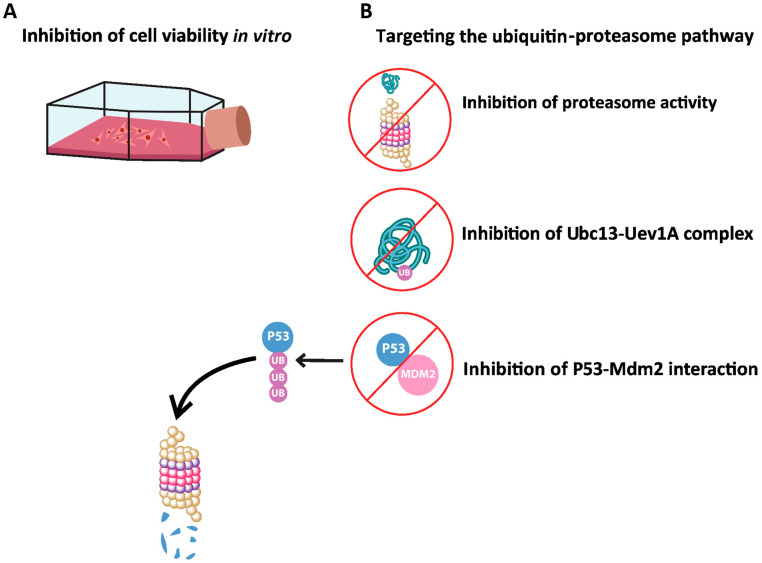
Anticancer effects of Halophila stipulacea. (**A**) Extracts from *H. stipulacea* inhibited the proliferation of several cell lines in vitro [54,55,57,60]. (**B**) Extracts from *H. stipulacea* inhibited proteasome activity and the interaction of Uev1A-Ubc13, which is associated with tumorigenesis. Also, the P53-Mdm2 interaction, which leads to p53 degradation, was inhibited by *H. stipulacea* extracts [55].

**Figure 5 biomolecules-14-00991-f005:**
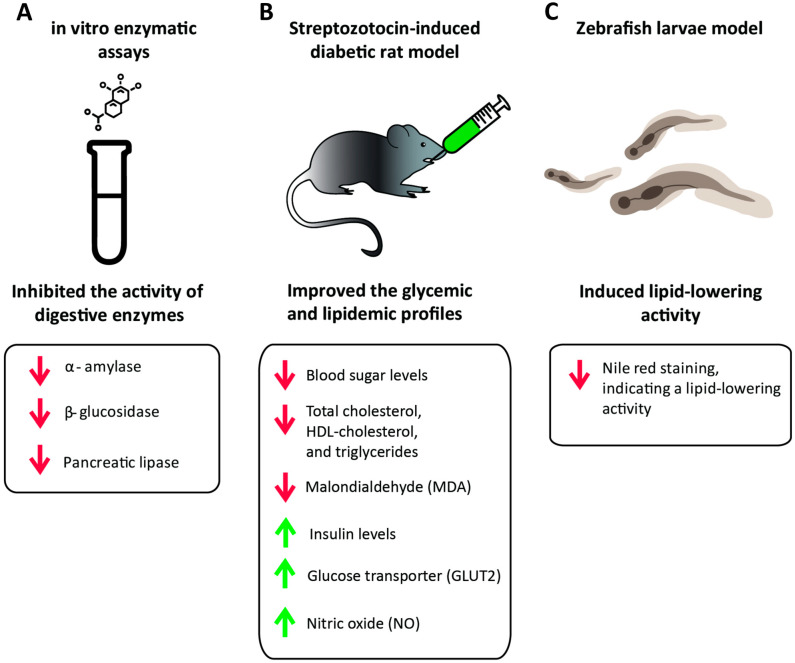
The effects of *Halophila stipulacea* on metabolic disorders. (**A**) An ethanolic extract from *H. stipulacea* strongly inhibited the digestive enzymes α-amylase, β-glucosidase, and pancreatic lipase [56]. (**B**) Oral administration of *H. stipulacea* ethanolic extract to a streptozotocin-induced diabetic rat model significantly reduced blood sugar, total cholesterol, HDL-cholesterol, triglycerides, and MDA levels, while increasing insulin, GLUT2, and NO levels [56]. (**C**) Ethyl acetate and methanolic extracts from *H. stipulacea* leaves resulted in a significant reduction in Nile red staining after 48 h of exposure, indicating a strong lipid-lowering activity [54].

**Table 1 biomolecules-14-00991-t001:** Taxonomy of *Halophila stipulacea*. Retrieved from: WoRMS (https://www.marinespecies.org/aphia.php?p=taxdetails&id=145714) (accessed on 22 June 2024).

Phylum	Tracheophyta
Class	Magnoliopsida
Order	Alismatales
Family	Hydrocharitaceae
Genus	*Halophila*
Species	*Halophila stipulacea*
Binomial name	*Halophila stipulacea* (Forsskål and Niebuhr) Ascherson

**Table 3 biomolecules-14-00991-t003:** Nutritional composition of *Halophila stipulacea* extracts.

Extract Type	Methods	Main Results	References
Dried *H. stipulacea* leaves powder	Biochemical analyses of ash, lipid, and protein	- The ash, lipid, and protein contents of *H. stipulacea* were 14.56 ± 2.08%, 3.16 ± 0.48%, and 8.11 ± 0.07%, respectively	[63]
Aqueous crude extract	HPGC	- The seagrass *H. stipulacea* was rich in galactose (20.5%), mannose (15.2%), and glucose (12.1%)	[60]
Dried *H. stipulacea* leaves powder	HPLC Pico-Tag	- The total protein content of *H. stipulacea* was 3.06% - The amino acid content of *H. stipulacea* was 28.6 mg/g dry weight - The seagrass *H. stipulacea* was rich in essential amino acids including valine, threonine, methionine, and leucine, with concentrations of 3.4, 2.7, 2.2, and 1.4 mg/g dry weight, respectively - The seagrass *H. stipulacea* was rich in the non-essential amino acids cysteine, tyrosine, and arginine, with concentrations of 4.1, 3.6, and 3.1 mg/g dry weight, respectively	[60]
Saponifiable matter (from the CHCl3 fraction of the MeOH/CHCl3 crude extract)	GC-MS	- The seagrass *H. stipulacea* contained high amounts of saturated fatty acids (56.63%), particularly palmitic acid (33.70%) and Pentadecanoic acid (10.25%) - The seagrass *H. stipulacea* contained high amounts of unsaturated fatty acids (28.21%), especially linoleic acid (14.94%), and α-linolenic acid (7.64%)	[60]
Dried *H. stipulacea* young leaves powder	Biochemical analyses of proteins, carbohydrates, and lipids	- Protein, carbohydrate, and lipid concentrations were 21.3 ± 0.5, 19.0 ± 0.6, and 5.8 ± 0.1 mg/g dry weight, respectively - The macromolecular content of young leaves was significantly greater than that of old leaves	[64]
Dried *H. stipulacea* old leaves powder	Biochemical analyses of proteins, carbohydrates, and lipids	- Protein, carbohydrate, and lipid concentrations were 8.2 ± 0.6, 10.9 ± 1.1, and 1.6 ± 0.3 mg/g dry weight, respectively	[64]
Dried *H. stipulacea* leaves powder	Wet digestion using nitric and perchloric acid	- Concentrations of Fe, Zn, and K were highest in summer and autumn - Concentrations of Zn, Na, K, and Mg were 25.4 ± 4.2, 62,596 ± 2410, 18,023 ± 563, and 11,788 ± 411 ug/g dry weight, respectively, which were higher compared to the stems, roots, and rhizoids	[65]
Dried *H. stipulacea* stems, roots, and rhizoids powder	Wet digestion using nitric and perchloric acid	- Concentrations of Zn, Na, K, and Mg were 20.1 ± 3.7, 61,170 ± 2862, 15,414 ± 1274, and 10,708 ± 317 ug/g dry weight	[65]

**Table 4 biomolecules-14-00991-t004:** The antimicrobial activity of *Halophila stipulacea* extracts.

Extract	Dose	Experimental Model	Organisms	Main Results	References
Anti-bacterial					
Ethanolic extract of leaves	200 mg/mL	- Method: Agar well diffusion - Positive control: Ticarcillin (25 μg), Cefepime (30 μg), Gentamicin (10 μg), Amikacin (30 μg), Imipenem (10 μg), Piperacillin (100 μg), Ampicillin (10 μg), Augmentin (30 μg), Cefoxltin (30 μg), Cephalothin (30 μg), Cotrimoxazole (25 μg)	*Bacillus subtilis*, Methicillin-Resistant *Staphylococcus aureus*, *Micrococcus luteus*, *Staphylococcus aureus*, *Escherichia coli*, *Klebsiella pneumonia*, *Pseudomonas aeruginosa*	- Strong antibacterial activity (higher than the other extracts) - Highest activity against *Pseudomonas aeruginosa*, with an inhibition zone of 20.67 ± 0.58 mm	[66]
Chloroform extract of leaves	200 mg/mL	- Method: Agar well diffusion - Positive control: Ticarcillin (25 μg), Cefepime (30 μg), Gentamicin (10 μg), Amikacin (30 μg), Imipenem (10 μg), Piperacillin (100 μg), Ampicillin (10 μg), Augmentin (30 μg), Cefoxltin (30 μg), Cephalothin (30 μg), Cotrimoxazole (25 μg)	*Bacillus subtilis*, Methicillin-Resistant *Staphylococcus aureus*, *Micrococcus luteus*, *Staphylococcus aureus*, *Escherichia coli*, *Klebsiella pneumonia*, *Pseudomonas aeruginosa*	- Strong antibacterial activity - Highest activity against *Pseudomonas aeruginosa*, with an inhibition zone of 18.33 ± 0.58 mm	[66]
Ethyl acetate extract of leaves	200 mg/mL	- Method: Agar well diffusion - Positive control: Ticarcillin (25 μg), Cefepime (30 μg), Gentamicin (10 μg), Amikacin (30 μg), Imipenem (10 μg), Piperacillin (100 μg), Ampicillin (10 μg), Augmentin (30 μg), Cefoxltin (30 μg), Cephalothin (30 μg), Cotrimoxazole (25 μg)	*Bacillus subtilis*, Methicillin-Resistant *Staphylococcus aureus*, *Micrococcus luteus*, *Staphylococcus aureus*, *Escherichia coli*, *Klebsiella pneumonia*, *Pseudomonas aeruginosa*	- Strong antibacterial activity - Highest activity against *Pseudomonas aeruginosa*, with an inhibition zone of 17.33 ± 0.58 mm	[66]
Aqueous extract of leaves	200 mg/mL	- Method: Agar well diffusion - Positive control: Ticarcillin (25 μg), Cefepime (30 μg), Gentamicin (10 μg), Amikacin (30 μg), Imipenem (10 μg), Piperacillin (100 μg), Ampicillin (10 μg), Augmentin (30 μg), Cefoxltin (30 μg), Cephalothin (30 μg), Cotrimoxazole (25 μg)	*Bacillus subtilis*, Methicillin-Resistant *Staphylococcus aureus*, *Micrococcus luteus*, *Staphylococcus aureus*, *Escherichia coli*, *Klebsiella pneumonia*, *Pseudomonas aeruginosa*	- Showed antibacterial activity only against *Pseudomonas aeruginosa* (15.67 ± 1.16 mm inhibition zone)	[66]
Ethanolic extract of leaves + Antibiotic	200 mg/mL	- Method: Agar well diffusion - Positive control: Ticarcillin (25 μg), Cefepime (30 μg), Gentamicin (10 μg), Amikacin (30 μg), Imipenem (10 μg), Piperacillin (100 μg), Ampicillin (10 μg), Augmentin (30 μg), Cefoxltin (30 μg), Cephalothin (30 μg), Cotrimoxazole (25 μg)	*Pseudomonas aeruginosa*	- Showed synergistic effect with Imipenem, Piperacillin, Cefoxltin, Cephalothin, and Cotrimoxazole, with inhibition zones of 38.5, 31.5, 30.5, 24.5, and 24.5 mm, respectively	[66]
Aqueous extract of the seagrass		- Method: Agar disk diffusion - Positive control: None	*Escherichia coli*, *Pseudomonas aeruginosa*, *Bacillus subtilis*	- Showed antibacterial activity only against Gram positive bacteria *Bacillus subtilis* (15 mm inhibition zone)	[67]
Methanolic extract of leaves	100 mg/mL	- Method: Agar disk diffusion - Positive control: streptomycin	*Staphylococcus aureus*, *Vibrio cholera*, *Shigella dysentriae*, *Shigella bodii*, *Salmonella paratyphi*, *Pseudomonas aeruginosa*, *Klebsiella pneumonia*,	- Showed strongest activity between all tested seagrasses against *Shigella dysentriae* with the lowest MIC value (100 µg/mL)	[68]
Chloroform extract of leaves	100 mg/mL	- Method: Agar disk diffusion - Positive control: streptomycin	*Staphylococcus aureus*, *Vibrio cholera*, *Shigella dysentriae*, *Shigella bodii*, *Salmonella paratyphi*, *Pseudomonas aeruginosa*, *Klebsiella pneumonia*,	- Strong activity against *Shigella bodii*, with an MIC value of 100 µg/mL	[68]
Hexane extract of leaves	100 mg/mL	- Method: Agar disk diffusion - Positive control: streptomycin	*Staphylococcus aureus*, *Vibrio cholera*, *Shigella dysentriae*, *Shigella bodii*, *Salmonella paratyphi*, *Pseudomonas aeruginosa*, *Klebsiella pneumonia*,	- Showed activity against all tested organisms except *Staphylococcus aureus* - Showed strong activity against *Shigella bodii* (4 mm inhibition zone)	[68]
Ethanolic extract	1 mg/mL	- Method: Agar disk diffusion - Positive control: Gentamycin (4 µg/mL) Amphotericin B (100 µg/mL)	*Micrococcus* sp., *Bacillus cereus*, *Enterococcus faecalis*, *Proteus vulgaris*, *Pseudomonasaeruginosa*, *Enterobacter cloacae*	- Showed strong effect on the gram-negative bacterium *Proteus vulgaris* (11 mm inhibition zone)	[61]
Hexane extract		- Method: Agar disk diffusion - Positive control: None	*Escherichia coli*, *Escherichia coli DH5 (alpha)*, *listeria monocytogenes*, *Salmonella enterica*, *Agrobacterium* *tumefaciens*, *Pseudomonas aerigunosa*, *Staphylococus aureus*, *Micrococcus luteus*	- Showed no significant effect on all tested bacteria	[69]
Ethyl acetate extract		- Method: Agar disk diffusion - Positive control: None	*Escherichia coli*, *Escherichia coli DH5 (alpha)*, *listeria monocytogenes*, *Salmonella enterica*, *Agrobacterium* *tumefaciens*, *Pseudomonas aerigunosa*, *Staphylococus aureus*, *Micrococcus luteus*	- Showed an inhibitory effect against *listeria monocytogenes* and *Salmonella enterica* (10 mm inhibition zone for both)	[69]
Methanolic extract		- Method: Agar disk diffusion - Positive control: None	*Escherichia coli*, *Escherichia coli DH5 (alpha)*, *listeria monocytogenes*, *Salmonella enterica*, *Agrobacterium* *tumefaciens*, *Pseudomonas aerigunosa*, *Staphylococus aureus*, *Micrococcus luteus*	- Showed a strong activity against *Staphylococus aureus* and *Micrococcus luteus* (inhibition zone between 10–15 mm for both)	[69]
Hexane extract	5 mg/mL	- Method: Agar disk diffusion - Positive control: None	*Pseudomonas aeruginosa*	- Showed moderate activity that was higher than other tested seagrasses (3 mm inhibition zone)	[70]
Chloroform extract	5 mg/mL	- Method: Agar disk diffusion - Positive control: None	*Pseudomonas aeruginosa*	- Showed moderate activity that was higher than other tested seagrasses (2 mm inhibition zone)	[70]
Methanolic extract	5 mg/mL	- Method: Agar disk diffusion - Positive control: None	*Pseudomonas aeruginosa*	- Showed the highest activity among other tested solvents (7 mm inhibition zone)	[70]
Fabric treated with *H. stipulacea* ethanolic extract		- Method: Antibacterial properties of finished fabrics - Positive control: None	*Staphylococcus aureus*, *Escherichia coli*	- Strong bacterial growth inhibition	[62]
Anti-fungal					
Aqueous extract of the seagrass		- Method: Agar disk diffusion - Positive control: None	*Aspergillus niger*, *Candida albicans*	- Strong antifungal activity - Inhibition was greater against the fungi *Aspergillus niger* (inhibition zone of 20 mm)	[67]
Ethanolic extract	1 mg/mL	- Method: Agar disk diffusion - Positive control: Amphotericin B (100 µg/mL)	*Grotricumcandidum*, *Syncephalastrum racemosum*, *Penicillium* *marneffeii*, *Cryptococcus neoformas*	- Showed no effects on all tested fungi	[61]
Hexane extract	10 mg/disk	- Method: Agar disk diffusion - Positive control: None	*Aspergilus niger*, *saccharomyces cerevisiae*, *Candida tropicalis*	- Showed no significant effect	[69]
Ethyl acetate extract	10 mg/disk	- Method: Agar disk diffusion - Positive control: None	*Aspergilus niger*, *saccharomyces cerevisiae*, *Candida tropicalis*	- Showed no significant effect	[69]
Methanolic extract	10 mg/disk	- Method: Agar disk diffusion - Positive control: None	*Aspergilus niger*, *saccharomyces cerevisiae*, *Candida tropicalis*	- Showed no significant effect	[69]
Hexane extract	5 mg/mL	- Method: Agar disk diffusion - Positive control: None	*Macrophomina phaseolina*, *Colletotrichum capsici*, *Fusarium* sp., *Aspergillus flavus*	- Strong antifungal activity against *Macrophomina phaseolina* and *Colletotrichum capsici* (7 mm inhibition zone for both)	[70]
Chloroform extract	5 mg/mL	- Method: Agar disk diffusion - Positive control: None	*Macrophomina phaseolina*, *Colletotrichum capsici*, *Fusarium* sp., *Aspergillus flavus*	- Strong antifungal activity against *Colletotrichum capsici*, with an inhibition zone of 10 mm	[70]
Methanolic extract	5 mg/mL	- Method: Agar disk diffusion - Positive control: None	*Macrophomina phaseolina*, *Colletotrichum capsici*, *Fusarium* sp., *Aspergillus flavus*	- Strong antifungal activity against *Aspergillus flavus*, *Macrophomina phaseolina* and *Colletotrichum capsici*, with inhibition zones of 11, 16, and 13 mm, respectively	[70]
Anti-biofouling					
Hexane extract of leaves	3 and 30 µg/mL	- Method: Antifouling activity bioassay against marine bacteria (microfouling) - Positive control: None	*Cobetia marina*, *Vibrio harveyi*, *Roseobacter litoralis*, *Halomonas aquamarina*, *Pseudoalteromonas atlantica*,	- Showed strongest inhibition against *Halomonas aquamarina* and *Pseudoalteromonas atlantica* (20% growth inhibition at 30 µg/mL against both bacteria)	[54]
Hexane extract of leaves	60 to 3.75 µg/mL	- Method: Antifouling activity bioassay against mussel larval settlement - (macrofouling) - Positive control: CuSO_4_ (5 µM)	Mussel larvae (*Mytilus galloprovincialis*)	- Moderate non-significant inhibition against the settlement of the plantigrade larvae at 30 µg/mL	[54]
Ethyl acetate extract of leaves	3 and 30 µg/mL	- Method: Antifouling activity bioassay against marine bacteria (microfouling) - Positive control: None	*Cobetia marina*, *Vibrio harveyi*, *Roseobacter litoralis*, *Halomonas aquamarina*, *Pseudoalteromonas atlantica*,	- Showed strongest inhibition against *Cobetia marina* (25% growth inhibition at 30 µg/mL)	[54]
Ethyl acetate extract of leaves	60 to 3.75 µg/mL	- Method: Antifouling activity bioassay against mussel larval settlement - (macrofouling) - Positive control: CuSO_4_ (5 µM)	Mussel larvae *(Mytilus galloprovincialis)*	- Moderate non-significant inhibition against the settlement of the plantigrade larvae at 30 µg/mL	[54]
Methanol extract of leaves	3 and 30 µg/mL	- Method: Antifouling activity bioassay against marine bacteria (microfouling) - Positive control: None	*Cobetia marina*, *Vibrio harveyi*, *Roseobacter litoralis*, *Halomonas aquamarina*, *Pseudoalteromonas atlantica*,	- Showed moderate inhibition against *Halomonas aquamarina* and *Cobetia marina*, with growth inhibition at 30 µg/mL of 20 and 15%, respectively	[54]
Methanol extract of leaves	60 to 3.75 µg/mL	- Method: Antifouling activity bioassay against mussel larval settlement - (macrofouling) - Positive control: CuSO_4_ (5 µM)	Mussel larvae *(Mytilus galloprovincialis)*	- Strong, significant inhibition against the settlement of the plantigrade larvae at 30 µg/mL - EC50 value: 17.5 µg/mL	[54]
Hexane extract of stems	3 and 30 µg/mL	- Method: Antifouling activity bioassay against marine bacteria (microfouling) - Positive control: None	*Cobetia marina*, *Vibrio harveyi*, *Roseobacter litoralis*, *Halomonas aquamarina*, *Pseudoalteromonas atlantica*,	- Showed moderate inhibition against *Halomonas aquamarina* and *Cobetia marina*, with growth inhibition at 30 µg/mL of 15 and 20%, respectively	[54]
Hexane extract of stems	60 to 3.75 µg/mL	- Method: Antifouling activity bioassay against mussel larval settlement - (macrofouling) - Positive control: CuSO_4_ (5 µM)	Mussel larvae *(Mytilus galloprovincialis)*	- Strong, significant inhibition against the settlement of the plantigrade larvae at 30 µg/mL - EC50 value: 11.3 µg/mL	[54]
Ethyl acetate extract of stems	3 and 30 µg/mL	- Method: Antifouling activity bioassay against marine bacteria (microfouling) - Positive control: None	*Cobetia marina*, *Vibrio harveyi*, *Roseobacter litoralis*, *Halomonas aquamarina*, *Pseudoalteromonas atlantica*,	- Showed moderate inhibition against *Halomonas aquamarina* and *Cobetia marina*, with growth inhibition at 30 µg/mL of 15 and 25%, respectively	[54]
Ethyl acetate extract of stems	60 to 3.75 µg/mL	- Method: Antifouling activity bioassay against mussel larval settlement - (macrofouling) - Positive control: CuSO_4_ (5 µM)	Mussel larvae *(Mytilus galloprovincialis)*	- Moderate non-significant inhibition against the settlement of the plantigrade larvae at 30 µg/mL	[54]
Methanol extract of stems	3 and 30 µg/mL	- Method: Antifouling activity bioassay against marine bacteria (microfouling) - Positive control: None	*Cobetia marina*, *Vibrio harveyi*, *Roseobacter litoralis*, *Halomonas aquamarina*, *Pseudoalteromonas atlantica*,	- Showed moderate inhibition against *Halomonas aquamarina*, *Cobetia marina*, and *Pseudoalteromonas atlantica*, with growth inhibition at 30 µg/mL of 5, 20, and 10%, respectively	[54]
Methanol extract of stems	60 to 3.75 µg/mL	- Method: Antifouling activity bioassay against mussel larval settlement - (macrofouling) - Positive control: CuSO_4_ (5 µM)	Mussel larvae (*Mytilus galloprovincialis*)	- Moderate non-significant inhibition against the settlement of the plantigrade larvae at 30 µg/mL	[54]

**Table 5 biomolecules-14-00991-t005:** The antioxidant activities of *Halophila stipulacea* extracts.

Extract	Dose	Methods	Observations	References
Ethanolic extract of leaves	0.1 mg/mL	- Total antioxidant activity	- The antioxidant activity of the extract was higher than 6 other species of tested seagrasses	[32]
Ethanolic extract of leaves	Not specified	- DPPH radical scavenging assay - Positive control: ascorbic acid, gallic acid	- The extract showed good antioxidant activity reaching 67% DPPH radical scavenging, higher than 6 species of the tested seagrasses	[32]
Ethanolic extract of leaves	Not specified	- FRAP assay - Positive control: ascorbic acid	- The extract showed the highest ability in ferric ion reduction compared to the other tested seagrasses	[32]
Methanolic extract of *H. stipulacea*	Not specified	- DPPH radical scavenging assay	- The extract showed moderate antioxidant activity reaching 40% DPPH radical scavenging	[67]
Aqueous crude extract	Not specified	- DPPH radical scavenging assay - Positive control: quercetin, gallic acid	- IC50 13.3 ± 0.25 µg/mL - The extract showed mild antioxidant activity	[60]
Phenolic fraction of the methanolic crude extract	Not specified	- DPPH radical scavenging assay - Positive control: quercetin, gallic acid	- IC50 12.6 ± 1.9 µg/mL - phenolic fraction showed mild antioxidant activity	[60]
Ethanolic extract of leaves	20, 40, 60, 80, and 100 µg/mL	- DPPH radical scavenging assay - Positive control: ascorbic acid	- The extract at 100 µg/mL had strong DPPH radical scavenging activity (79%)	[66]
Ethanolic extract of old leaves	1, 10, and 100 µg/mL	- DPPH radical scavenging assay	- The extract at 100 µg/mL displayed 85% inhibition of the DPPH radical	[64]
Ethanolic extract of young leaves	1, 10, and 100 µg/mL	- DPPH radical scavenging assay	- The extract at 100 µg/mL displayed 45% inhibition of the DPPH radical, showing moderate activity	[64]
Ethanolic extract of old leaves	1, 10, and 100 µg/mL	- H_2_O_2_ damage protection assay	- Pre-treatment at 100 µg/mL increased the metabolically active WI38 cells percentage to 109%, showing strong protective activity	[64]
Ethanolic extract of young leaves	1, 10, and 100 µg/mL	- H_2_O_2_ damage protection assay	- Pre-treatment at 100 µg/mL increased the metabolically active WI38 cells percentage to 81%, showing good protective activity	[64]
Ethanolic extract of old leaves	1, 10, and 100 µg/mL	- Cell viability recovery assay	- The metabolically active cell percent of the injured WI-38 cells increased to 100% at 100 µg/mL	[64]
Ethanolic extract of young leaves	1, 10, and 100 µg/mL	- Cell viability recovery assay	- The metabolically active cell percent of the injured WI-38 cells increased to 83% at 100 µg/mL	[64]
Ethanolic extract of old leaves	10 μg/mL	- PCR Array analysis of oxidative stress defense genes	- Pre-treatment of WI-38 cells upregulated oxidative stress defense genes, such as GPX5, KRT1, LPO, MT3, NOX5, and TPO - Treatment of WI-38 cells after injury significantly upregulated GPX5, KRT1, LPO, MT3, NOX5, and TPO - Results showed that the extract could be more therapeutic than preventive against oxidative damage	[64]

**Table 6 biomolecules-14-00991-t006:** The anticancer activities of *Halophila stipulacea* extracts.

Extract	Dose	Methods	Observations	References
Hexane extract of leaves	10 or 30 µg/mL EC50 (µg/mL): - 19.5 ± 5.8 (on HCT116 for 24 h) - 29.1 ± 7.5 (on HCT116 for 48 h) - >30 (on hCMEC for 24 h) - >30 (on hCMEC for 48 h)	- In vitro MTT cell viability assay on human osteosarcoma (MG-63), human neuroblastoma (SHSY5Y), human colorectal carcinoma (HCT116), and normal cells (hCMEC) - 3D cell culture of human colorectal carcinoma (HCT-116) Cell Line	- Inhibited the viability of SHSY5Y and HCT116 cell lines after the treatment at 30 µg/mL for 48 h by approximately 50%. - Caused a slight increase in the number of dead cells in the 3D cell culture model	[54]
Ethyl acetate extract of leaves	10 or 30 µg/mL EC50 (µg/mL): - >30 (on MG-63 for 24 h) - 29.4 ± 6.3 (on MG-63 for 48 h) - 10.6 ± 7.0 (on SHSY5Y for 24 h) - 15.2 ± 1.9 (on SHSY5Y for 48 h) - 11.3 ± 1.8 (on hCMEC for 24 h) - 24.5 ± 15.6 (on hCMEC for 48 h)	- In vitro MTT cell viability assay on human osteosarcoma (MG-63), human neuroblastoma (SHSY5Y), human colorectal carcinoma (HCT116), and normal cells (hCMEC) - 3D cell culture of human colorectal carcinoma (HCT-116) Cell Line	- Inhibited the viability of MG-63, SHSY5Y, and HCT116 cell lines after the treatment at 30 µg/mL for 48 h by approximately 50%. - No significant effect on 3D cell culture model	[54]
Methanol extract of leaves	10 or 30 µg/mL	- In vitro MTT cell viability assay on human osteosarcoma (MG-63), human neuroblastoma (SHSY5Y), human colorectal carcinoma (HCT116), and normal cells (hCMEC) - 3D cell culture of human colorectal carcinoma (HCT-116) Cell Line	- No significant inhibition of cell proliferation - No significant effect on 3D cell culture model	[54]
Hexane extract of stems	10 or 30 µg/mL EC50 (µg/mL): - 7.6 ± 5.4 (on HCT116 for 24 h) - 25.4 ± 4.2 (on HCT116 for 48 h) - >30 (on hCMEC for 24 h) - >30 (on hCMEC for 48 h)	- In vitro MTT cell viability assay on human osteosarcoma (MG-63), human neuroblastoma (SHSY5Y), human colorectal carcinoma (HCT116), and normal cells (hCMEC) - 3D cell culture of human colorectal carcinoma (HCT-116) Cell Line	- Inhibited the viability of SHSY5Y and HCT116 cell lines after the treatment at 30 µg/mL for 48 h by approximately 50%. - Showed a toxicity level in cancer cell lines that was four times greater than that in the normal cell line after 24 h of treatment. - No significant effect on 3D cell culture model	[54]
Ethyl acetate extract of stems	10 or 30 µg/mL >EC50 (µg/mL): - >30 (on MG-63 for 24 h) - 19.1 ± 9.0 (on MG-63 for 48 h) - 23.4 ± 1.1 (on SHSY5Y for 24 h) - 18.7 ± 3.1 (on SHSY5Y for 48 h) - 9.2 ± 0.2 (on hCMEC for 24 h) - 15.4 ± 1.1 (on hCMEC for 48 h)	- In vitro MTT cell viability assay on human osteosarcoma (MG-63), human neuroblastoma (SHSY5Y), human colorectal carcinoma (HCT116), and normal cells (hCMEC) - 3D cell culture of human colorectal carcinoma (HCT-116) Cell Line	- Inhibited the viability of MG-63, SHSY5Y, and HCT116 cell lines after the treatment at 30 µg/mL for 48 h by approximately 50%. - No significant effect on 3D cell culture model	[54]
Methanol extract of stems	10 or 30 µg/mL	- In vitro MTT cell viability assay on human osteosarcoma (MG-63), human neuroblastoma (SHSY5Y), human colorectal carcinoma (HCT116), and normal cells (hCMEC) - 3D cell culture of human colorectal carcinoma (HCT-116) Cell Line	- No significant inhibition of cell proliferation - No significant effect on 3D cell culture model	[54]
Aqueous crude extract	100 and 1000 µg/mL IC50 (µg/mL): - 10.70 (on DU-145) - 12.95 (on PANC-1)	- In vitro SRB cell viability assay on human ovarian cancer (SKOV-3), breast cancer (MCF-7), cervical cancer (HeLa), prostate cancer (DU-145), and pancreatic cancer (PANC-1) cell lines.	- Showed strong cytotoxic effect against prostate (DU-145), and pancreatic (PANC-1) cancer cell lines.	[60]
CHCl3 fraction of the MeOH/CHCl3 crude extract	100 and 1000 µg/mL IC50 (µg/mL): - 21.90 (on DU-145) - 6.40 (on HeLa) - 14.90 (on PANC-1)	- In vitro SRB cell viability assay on human ovarian cancer (SKOV-3), breast cancer (MCF-7), cervical cancer (HeLa), prostate cancer (DU-145), and pancreatic cancer (PANC-1) cell lines.	- Showed strong cytotoxic effect against prostate (DU-145), cervical (HeLa), and pancreatic (PANC-1) cancer cell lines.	[60]
Unsaponifiable matter (from the CHCl3 fraction of the MeOH/CHCl3 crude extract)	100 and 1000 µg/mL IC50 (µg/mL): - 14.0 (on MCF-7)	- In vitro SRB cell viability assay on human ovarian cancer (SKOV-3), breast cancer (MCF-7), cervical cancer (HeLa), prostate cancer (DU-145), and pancreatic cancer (PANC-1) cell lines.	- Showed strong cytotoxic effect against breast cancer (MCF-7) cell line.	[60]
Phenolic fraction of the methanolic crude extract	100 and 1000 µg/mL	- In vitro SRB cell viability assay on human ovarian cancer (SKOV-3), breast cancer (MCF-7), cervical cancer (HeLa), prostate cancer (DU-145), and pancreatic cancer (PANC-1) cell lines.		[60]
Ethyl acetate fraction of the methanolic crude extract	5, 12.5, 25, and 50 μg/well IC50 (μg/mL): - HepG2: 4.73 - HCT116: 11.30 - MCF7: 10.30 - HeLa: 17.50	- In vitro SRB cell viability assay on human liver cancer (HepG2), colon cancer (HCT-116), breast cancer (MCF-7), and cervical cancer (HeLa) cell lines	- Showed the strongest inhibition against HepG2 liver cancer cell line	[55]
	20 μg/mL	- Proteasome activity assay	- The extract caused 50% inhibition	
	10 μg/mL	- Ubc13–Uev1A interaction inhibition assay	- The extract caused 55% inhibition	
	20 μg/mL	- P53-Mdm2 interaction inhibition assay	- The binding percentage of the extract was 32%	
Aqueous fraction of the methanolic crude extract	5, 12.5, 25, and 50 μg/well	- In vitro SRB cell viability assay on human liver cancer (HepG2), colon cancer (HCT-116), breast cancer (MCF-7), and cervical cancer (HeLa) cell lines		[55]
	20 μg/mL	- Proteasome activity assay	- The extract caused 61% inhibition	
	10 μg/mL	- Ubc13–Uev1A interaction inhibition assay	- The extract caused 62% inhibition	
	20 μg/mL	- P53-Mdm2 interaction inhibition assay	- The binding percentage of the extract was 57%	
Diethyl ether fraction of the acetone crude extract	5, 12.5, 25, and 50 μg/well IC50 (μg/mL): - HepG2: 3.98 - HCT116: 27.10 - MCF7: 5.33	- In vitro SRB cell viability assay on human liver cancer (HepG2), colon cancer (HCT-116), breast cancer (MCF-7), and cervical cancer (HeLa) cell lines	- Showed the strongest inhibition against HepG2 liver cancer and MCF-7 breast cancer cell lines	[55]
	20 μg/mL	- Proteasome activity assay	- The extract caused 97% inhibition	
	10 μg/mL	- Ubc13–Uev1A interaction inhibition assay	- The extract caused 62% inhibition	
	20 μg/mL	- P53-Mdm2 interaction inhibition assay	- The binding percentage of the extract was 67%	
Butanol fraction of the acetone crude extract	5, 12.5, 25, and 50 μg/well IC50 (μg/mL): - HepG2: 11.00 - MCF7: 9.98	- In vitro SRB cell viability assay on human liver cancer (HepG2), colon cancer (HCT-116), breast cancer (MCF-7), and cervical cancer (HeLa) cell lines	- Showed the strongest inhibition against MCF-7 breast cancer cell line	[55]
	20 μg/mL	- Proteasome activity assay	- The extract caused 86% inhibition	
	10 μg/mL	- Ubc13–Uev1A interaction inhibition assay	- The extract caused 56% inhibition	
	20 μg/mL	- P53-Mdm2 interaction inhibition assay	- The binding percentage of the extract was 67%	
Ethanolic extract of roots and shoots	156.25, 312.5, 625, 1250, 2500, 5000, and 10,000 µg/mL	- In vitro MTT cell viability assay on human pancreatic cancer (PA1), lung cancer (A549), prostate cancer (PC3), and colon cancer (Caco2) cell lines	- The extract showed the lowest anticancer activity compared to the other tested plant extracts	[57]

**Table 7 biomolecules-14-00991-t007:** The anti-metabolic disorders activities of *Halophila stipulacea* extracts.

Extract	Dose	Methods	Observations	References
Ethanolic extract	- 600 µg/mL - IC50: 250.62 ± 8.2 µg/mL	- α-amylase inhibition assay - Positive control: Acarbose	- Strong enzymatic inhibition possibly due to the high level of phenolic compounds	[56]
Ethanolic extract	- 300 µg/mL - IC50: 380 ± 3.5 µg/mL	- β-glucosidase inhibition assay - Positive control: Acarbose	- Strong enzymatic inhibition possibly due to the high level of phenolic compounds	[56]
Ethanolic extract	- 100 µg/mL - IC50: 23.05 ± 3.5 µg/mL	- Pancreatic lipase inhibition assay - Positive control: Orlistat	- Strong enzymatic inhibition possibly due to the high level of phenolic compounds	[56]
Ethanolic extract	100 and 200 mg/kg	- Streptozotocin-induced diabetic rat model - Positive control: glibenclamide (6.5 mg/kg)	- Decreased blood sugar levels and increased the insulin levels in a dose-dependent manner - Increased the levels of the glucose transporter GLUT2 and nitric oxide (NO) - Decreased the levels of total cholesterol, HDL-cholesterol, and triglycerides - Decreased the level of malondialdehyde (MDA), an indicator of free radical-induced lipid peroxidation	[56]
Hexane extract of leaves	10 and 30 µg/mL	- Glucose uptake assay (2-NBDG in HepG2 cells (followed by MTT assay)) - Positive control: Emodin (10 µM) - Method: SRB anti-steatosis assay	- No significant effect on the glucose uptake of HepG2 cells - No significant anti-steatosis effect on the fatty acid-overloaded HepG2 cells	[54]
Ethyl acetate extract of leaves	10 and 30 µg/mL	- Glucose uptake assay (2-NBDG in HepG2 cells (followed by MTT assay)) - Positive control: Emodin (10 µM) - SRB anti-steatosis assay	- No significant effect on the glucose uptake of HepG2 cells - No significant anti-steatosis effect on the fatty acid-overloaded HepG2 cells	[54]
Methanol extract of leaves	10 and 30 µg/mL	- Glucose uptake assay (2-NBDG in HepG2 cells (followed by MTT assay)) - Positive control: Emodin (10 µM) - SRB anti-steatosis assay	- No significant effect on the glucose uptake of HepG2 cells - No significant anti-steatosis effect on the fatty acid-overloaded HepG2 cells	[54]
Hexane extract of stems	10 and 30 µg/mL	- Glucose uptake assay (2-NBDG in HepG2 cells (followed by MTT assay)) - Positive control: Emodin (10 µM) - SRB anti-steatosis assay	- No significant effect on the glucose uptake of HepG2 cells - No significant anti-steatosis effect on the fatty acid-overloaded HepG2 cells	[54]
Ethyl acetate extract of stems	10 and 30 µg/mL	- Glucose uptake assay (2-NBDG in HepG2 cells (followed by MTT assay)) - Positive control: Emodin (10 µM) - SRB anti-steatosis assay	- No significant effect on the glucose uptake of HepG2 cells - No significant anti-steatosis effect on the fatty acid-overloaded HepG2 cells	[54]
Methanol extract of stems	10 and 30 µg/mL	- Glucose uptake assay (2-NBDG in HepG2 cells (followed by MTT assay)) - Positive control: Emodin (10 µM) - SRB anti-steatosis assay	- No significant effect on the glucose uptake of HepG2 cells - No significant anti-steatosis effect on the fatty acid-overloaded HepG2 cells	[54]
Hexane extract of leaves	2 and 6 µg/mL	- Zebrafish larvae Nile red fat metabolism assay - Positive control: resveratrol (50 µM)	- Decreased the Nile red staining in the treated zebrafish larvae	[54]
Ethyl acetate extract of leaves	- 2 and 6 µg/mL - EC50: 2.2 µg/mL	- Zebrafish larvae Nile red fat metabolism assay - Positive control: resveratrol (50 µM)	- Significantly reduced the Nile red staining in the treated zebrafish larvae, with an EC50 of 2.2 µg/mL	[54]
Methanol extract of leaves	- 2 and 6 µg/mL - EC50: 1.2 µg/mL	- Zebrafish larvae Nile red fat metabolism assay - Positive control: resveratrol (50 µM)	- Significantly reduced the Nile red staining in the treated zebrafish larvae, with an EC50 of 1.2 µg/mL	[54]
Hexane extract of stems	2 and 6 µg/mL	- Zebrafish larvae Nile red fat metabolism assay - Positive control: resveratrol (50 µM)	- Decreased the Nile red staining in the treated zebrafish larvae	[54]
Ethyl acetate extract of stems	2 and 6 µg/mL	- Zebrafish larvae Nile red fat metabolism assay - Positive control: resveratrol (50 µM)	- Decreased the Nile red staining in the treated zebrafish larvae	[54]
Methanol extract of stems	2 and 6 µg/mL	- Zebrafish larvae Nile red fat metabolism assay - Positive control: resveratrol (50 µM)	- Decreased the Nile red staining in the treated zebrafish larvae	[54]

**Table 8 biomolecules-14-00991-t008:** The anti-osteoclastogenic activities of *Halophila stipulacea* extracts.

Extract	Dose	Methods	Observations	References
Ethyl acetate fraction of the methanolic crude extract	100 μg/mL	- Method: tartrate-resistant acid - phosphatase (TRAP) assay on RAW264 cells - Positive control: Quercetin (100 μg/mL)	- Showed 17% inhibition of osteoclastogenesis.	[55]
Aqueous fraction of the methanolic crude extract	100 μg/mL	- Method: tartrate-resistant acid - phosphatase (TRAP) assay on RAW264 cells - Positive control: Quercetin (100 μg/mL)	- Showed 29% inhibition of osteoclastogenesis.	[55]
Diethyl ether fraction of the acetone crude extract	100 μg/mL	- Method: tartrate-resistant acid - phosphatase (TRAP) assay on RAW264 cells - Positive control: Quercetin (100 μg/mL)	- Showed the strongest antiosteoclastogenic activity among other tested extracts reaching 117% inhibition.	[55]
Butanol fraction of the acetone crude extract	100 μg/mL	- Method: tartrate-resistant acid - phosphatase (TRAP) assay on RAW264 cells - Positive control: Quercetin (100 μg/mL)	- Showed the second highest antiosteoclastogenic activity among other tested extracts reaching 114% inhibition.	[55]

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
