# Peer review of "Halophila stipulacea: A Comprehensive Review of Its Phytochemical Composition and Pharmacological Activities"

_biomolecules, 2024, doi:10.3390/biom14080991_

Round 1

Reviewer 1 Report

Comments and Suggestions for Authors

The present study titled “Halophila stipulacea: a comprehensive review of its phytochemical composition and pharmacological activities. The quality of the writing is good. The content of the article is informative. But authors should consider all section to increase the quality of manuscript.

-Please include appropriate references for Table 1.

-Do you have permission to use the images for Figure 1.

-Please increase the quality of Figure 2.

-Please use the same citation style for all text. (e.g Line 135-137)

-To summarize the nutritional composition of  H. stipulace, authors should add a table with appropriate references.

-You should include appropriate references for all figures.

-For figure 3, author summarized the studies related to health effects of H. stipulace, but in this way it is not clear which research studies found the results. Update the figure to show this.

- All abbreviations and words that need to be italicized in the text should be checked (e.g. 163)

-In table 3, please include quantitative data for the study results. For example, “Showed a strong activity against S. aureus and M. luteus” is not enough to understand results of experiments and compare them.

Author Response

Dear Reviewer,

Thank you for your constructive comments that help us improving the review during the revision.

Please find below our answers.

Regards

Dr M Maresca

Reviewer 1

The present study titled “Halophila stipulacea: a comprehensive review of its phytochemical composition and pharmacological activities. The quality of the writing is good. The content of the article is informative. But authors should consider all section to increase the quality of manuscript.

-Please include appropriate references for Table 1.

Thank you for your comment. We have included the appropriate reference for Table 1 in the caption.                

-Do you have permission to use the images for Figure 1.

Yes, we have the permission to use the images of Figure 1, and we have included the link to the source as well.

-Please increase the quality of Figure 2.

Thank you for your feedback. We have redrawn the chemical structures of Figure 2. with Chemdraw.

-Please use the same citation style for all text. (e.g Line 135-137)

The citation style has been updated to be consistent throughout the text.

-To summarize the nutritional composition of  H. stipulace, authors should add a table with appropriate references.

Thank you for your feedback. Table 3. has been added to summarize the nutritional composition of H. stipulacea.

-You should include appropriate references for all figures.

Thank you for your comment. The corresponding references are now included in all figures in the document.

-For figure 3, author summarized the studies related to health effects of H. stipulace, but in this way it is not clear which research studies found the results. Update the figure to show this.

Figure 3. has been updated to include the references.

- All abbreviations and words that need to be italicized in the text should be checked (e.g. 163)

The text has been revised accordingly.

-In table 3, please include quantitative data for the study results. For example, “Showed a strong activity against S. aureus and M. luteus” is not enough to understand results of experiments and compare them.

The table of antimicrobial activities (Table 4. now) has been updated to include quantitative data.

Reviewer 2 Report

Comments and Suggestions for Authors

Review papers always receive a lot of attention as they serve as a platform for further work in a given subject or field. The manuscript reviews the literature on the phytochemical content and pharmacological activity of Halophila stipulacea. 

A few comments that would help to better understand the information presented.

I suggest you reconsider and revised the order of sections and subsections of the manuscript and their titles.

It is not clear what the name of the section 3"General Characteristics" means, if the subsections are defined in terms of different subjects. Please revise the section structure.

The description of biochemical variations of plant in response to different environmental factors does not acceptable for subsection “Botanical characteristics”.

I would suggest subsection 3.1. specify as a “Botanical classification”. The source of the classification must be specified.

The information presented in the Table 2 must be described and discussed accordingly.

The lines   127-131 must be linked to the information presented and discussed in the Table 2

Please revise the first sentence of the section 3.3.2. to relate to the nutritional properties discussed below.

Provide an explanation of what you present and discuss as biological or pharmacological activity in the submitted manuscript, including the section 4.

Please do not use multiple activities in one Table 4. The information should be presented in different tables, in the relevant subsections. Table 4 should be modified only for antioxidant activity.

The Figs 4 and 5 should be explained. What are the sources of information?

Finally, a summary of the information or conclusions should be provided. A survey of previous work should generate new insights which are missing in present work.

 Some minor errors and inaccuracies:

The title of Table 3 should include “extrcts of Halophila stipulacea“.

Do not start the sentence with the conjunction "and" (43 L); It is a Lessepsian migrant…” (what “is it”?)… (49 L). Do not start a sentence with an abbreviation like H. stipulacea (66, 76, 98, 104, 328 etc.).

Do not use the capital letters in chemicals names like “Galic acid”, 2,2-Diphenyl-1-picrylhydrazyl“.

Genus and species name please use in italics (Table 1; 125, 167, 171, 387 etc.).

Author Response

Dear Reviewer,

Thank you for your comments that helped us to improve the revised version of our review.

Please find below our answers.

Regards

Dr M Maresca

Reviewer 2

Review papers always receive a lot of attention as they serve as a platform for further work in a given subject or field. The manuscript reviews the literature on the phytochemical content and pharmacological activity of Halophila stipulacea.

A few comments that would help to better understand the information presented.

I suggest you reconsider and revised the order of sections and subsections of the manuscript and their titles.

It is not clear what the name of the section 3"General Characteristics" means, if the subsections are defined in terms of different subjects. Please revise the section structure.

Thank you for your comment. The section structure has been revised.

The description of biochemical variations of plant in response to different environmental factors does not acceptable for subsection “Botanical characteristics”.

Thank you for your feedback. We have relocated the description of biochemical variations in response to environmental factors to section 3.2. Ecological Characteristics.

I would suggest subsection 3.1. specify as a “Botanical classification”. The source of the classification must be specified.

Thank you for your feedback. We have changed the section title to 3. Taxonomic classification of Halophila stipulacea. We have included the reference (source of the classification) in the caption of Table 1. and in the text.

The information presented in the Table 2 must be described and discussed accordingly.

The lines 127-131 must be linked to the information presented and discussed in the Table 2

The text has been updated, and further discussion of Table 2. has been added.

Please revise the first sentence of the section 3.3.2. to relate to the nutritional properties discussed below.

Thank you for your feedback. The first sentence of section 3.3.2 was changed.

Provide an explanation of what you present and discuss as biological or pharmacological activity in the submitted manuscript, including the section 4.

Thank you for your comment. This has been addressed in the introduction of this section.

Please do not use multiple activities in one Table 4. The information should be presented in different tables, in the relevant subsections. Table 4 should be modified only for antioxidant activity.

Thank you for your feedback. Table 4 (Table 5. now) has been updated and is now limited to the antioxidant activity only as suggested. Other activities were separated into different tables and placed in the relevant subsections.

The Figs 4 and 5 should be explained. What are the sources of information?

Thank you for your feedback. We have provided explanations for Figures 4 and 5 in their captions. Also, corresponding references have been added.

Finally, a summary of the information or conclusions should be provided. A survey of previous work should generate new insights which are missing in present work.

Thank you for your feedback. We have added a new section: 6. Conclusion and Future Perspectives.

 Some minor errors and inaccuracies:

The title of Table 3 should include “extrcts of Halophila stipulacea“.

We have updated the title of the table to include “extracts”.

Do not start the sentence with the conjunction "and" (43 L); „It is a Lessepsian migrant…” (what “is it”?)… (49 L). Do not start a sentence with an abbreviation like H. stipulacea (66, 76, 98, 104, 328 etc.).

The text has been revised

Do not use the capital letters in chemicals names like “Galic acid”, 2,2-Diphenyl-1-picrylhydrazyl“.

The text has been revised

Genus and species name please use in italics (Table 1; 125, 167, 171, 387 etc.).

The text has been revised

Round 2

Reviewer 2 Report

Comments and Suggestions for Authors

The Authors made substantial changes to the structure of the manuscript and clarified the description of the seagrass results review.

Please correct some inaccuracies:

In Table 4 and 5, column“Main results“ please use full latin name of microbs in all positions (like P. vulgaris, l. monocytognes, S. enteric etc.?).

Please clarify and unify the presented information in Table 4 (column „Experimental model“).

I would suggest table 7 to be deleted, because the information of three positions of one cited source is repeated, column "Dose" has no information at all. Please describe the missing information in the text of the subsection 6.4.

In the Table 8 and partilly in Table 5, the name of the column "Methods" indicates the content of the information provided, so do not repeat the word "method" for specific methods.

"In virto" please write in italics (Table 6 etc.)

Author Response

Dear Reviewer,

Thank you for your valuable comments that helped us to improve further our review.

Please find below our answers to your comments

Regards

Dr M Maresca

Reviewer

The Authors made substantial changes to the structure of the manuscript and clarified the description of the seagrass results review.

Please correct some inaccuracies:

In Table 4 and 5, column“Main results“ please use full latin name of microbs in all positions (like P. vulgaris, l. monocytognes, S. enteric etc.?).

Thank you for your comment. We have addressed the issue and we have used the full Latin names of all microbes.

Please clarify and unify the presented information in Table 4 (column „Experimental model“).

Thank you for your comment. We have updated the "Experimental Model" column of Table 4. It is now consistent and includes the method and positive control used.

I would suggest table 7 to be deleted, because the information of three positions of one cited source is repeated, column "Dose" has no information at all. Please describe the missing information in the text of the subsection 6.4.

Thank you for your comment. We have removed Table 7. as suggested, and the text of subsection 6.4 has been updated.

In the Table 8 and partilly in Table 5, the name of the column "Methods" indicates the content of the information provided, so do not repeat the word "method" for specific methods.

Thank you for your feedback. We have updated Table 5. and Table 8. (Table 7. now).

"In virto" please write in italics (Table 6 etc.)

Thank you for your comment. This has been addressed.